

# Biological maturation determines the beneficial effects of high-intensity functional training on cardiorespiratory fitness in male adolescents

Jarosław Domaradzki[1], Cristian Alvarez[2], Rafał Szafraniec[1] and Dawid Koźlenia[1]

[1] Faculty of Physical Education and Sport, Wroclaw University of Health and Sport Sciences, Wrocław, Poland
[2] Exercise and Rehabilitation Sciences Institute, School of Physical Therapy, Faculty of Rehabilitation Sciences, Universidad Andres Bello, Santiago, Chile

Corresponding author
Jarosław Domaradzki,
jaroslaw.domaradzki@awf.wroc.pl

## ABSTRACT

**Background:** During adolescence, between 15–18 years of age, biological maturation is associated with changes in body composition, blood pressure (BP), and cardiorespiratory fitness (CRF). Also, environmental factors may influence the trajectory of these changes. The rising prevalence of physical inactivity calls for the identification of effective exercise modalities to mitigate adverse health outcomes in youth. Although high-intensity training regimens are increasingly recognized for their benefits in school settings, the specific role of biological maturation in determining adolescent responses to such intervention in health outcomes remains insufficiently understood.

**Aim:** This quasi-experimental study investigated the effects of chronological age (CA) and maturity offset (MO), defined as the time before/after peak height velocity, on changes in body fat (BF), BP, and CRF induced by high-intensity functional training (HIFT) and assessed the contribution of MO on these outcomes.

**Methods:** The sample consisted of 116 males, grouped by CA years (y) (15y [$n = 30$], 16y [$n = 29$], 17y [$n = 30$], 18y [$n = 27$]). The biological maturation effects were studied after separating the participants by quartiles of MO into four classes: late, middle late, middle early, and early matured. Participants were randomly allocated to experimental (EG) and control groups (CG). The EG performed HIFT for 8 weeks (6–14 min/twice a week). Changes in BF, BP, and CRF before and after the intervention were calculated. The MO role in the HIFT intervention was tested by MANCOVA and ANCOVA with *post hoc* detailed comparisons.

**Results:** MO contributed to the studied model more than CA (Wilk's $\Lambda = 0.49$, $\eta_p^2 = 0.24$, $p < 0.001$; Wilk's $\Lambda = 0.83$, $\eta_p^2 = 0.07$, $p < 0.063$, respectively). When controlling for age, MO was the main determinant (in comparison to CA) of the delta in CRF ($F = 8.76$, $p < 0.001$). Those who matured earliest (MO > 3.59 years after APHV) benefited more from HIFT intervention than biologically younger, independent of CA (improvement by $194.7 \pm 151.5$ m in the CRF test). Maturity offset was the primary contributor to the variance in $\Delta$CRF ($\beta = 0.71$, $r^2$sp = 9.3%, $p = 0.014$). When MO was combined with CA, both independent variables together explained 15% of the variance ($p = 0.004$).

**Conclusion:** Biological maturation plays a more important role than CA in determining HIFT-induced improvements in CRF among male adolescents: those who matured earliest exhibited the greatest gains, whereas changes in BP and BF were not significantly related to either factor. Tailoring HIFT-based interventions can be achieved by adjusting intensity, progression, and load according to each one maturity level, ensuring more developmentally appropriate exercise programs. Future research should investigate the feasibility of these maturity-focused strategies in broader adolescent populations, thereby informing larger-scale interventions and maximizing long-term health benefits.

## INTRODUCTION

The World Health Organization recommends at least 60 min of moderate to vigorous physical activity per day for adolescents, but only 20% of adolescents worldwide meet this recommendation (*Rocliffe et al., 2023*). Depending on the method used to assess exercise intensity, high intensity is indicated by ≥85% of $VO_2$max (*Kolnes et al., 2021*), >80% of the age-predicted maximum heart rate, and a subjective rating of at least 8 on a 10-point scale for perceived exertion (REP) (*Duncombe et al., 2024*). Given that the problem of insufficient physical activity is very common, it is imperative to explore strategies that encourage engagement in regular exercise. Physical education (PE) classes play a particularly critical role in this regard. High-intensity exercise can be used during PE classes in the form of high-intensity interval training (HIIT) (*Duncombe et al., 2024*), which may be defined as alternating periods of moderate- and high-intensity exercise (*Coates et al., 2023*). There are several variations of HIIT, one of which is high-intensity functional training (HIFT) (*Feito et al., 2018a*). HIFT emphasizes functional, multi-joint movements, incorporating both aerobic and muscle-strengthening exercises (*Heinrich et al., 2015*; *Teixeira et al., 2023*). *Eather, Morgan & Lubans (2016)* observed after an 8-week HIFT intervention significant increases in cardiovascular endurance (+10 laps in shuttle running), flexibility (+3 cm in the sit and reach test), and muscular fitness (+13 cm in the standing long jump). The added value may be greater enjoyment levels, as indicated by higher scores on the Physical Activity Enjoyment Scale ($p = 0.014$; d = 0.92) in participants of HIFT compared to those engaged in vigorous-intensity continuous training (*Mayr-Ojeda et al., 2022*). This is particularly important for the effectiveness of intervention programs delivered in the PE classes. On the other hand, the implementation of HIFT in physical education classes allows for the regular, controlled, and widespread introduction of high-intensity exercise into the lives of young people (*Popowczak et al., 2022*; *Domaradzki et al., 2021*; *Feito et al., 2018b*). Physical activity during PE classes has been demonstrated to have a significant association with the level of health-related fitness (H-RF) (*Chen et al., 2018*).

Many studies have evaluated the influence of sex, chronological age, and various interventions on health-related fitness (H-RF) outcomes (*Raghuveer et al., 2020*). However, biological maturation is a key developmental factor that may significantly moderate the effectiveness of physical activity (PA) interventions, especially in adolescents. While chronological age is often used as a proxy for development, it does not adequately reflect the physiological and hormonal changes that occur during puberty. These changes, including variations in growth rate, hormonal profiles, and neuromuscular development, can substantially influence how individuals respond to exercise interventions (*Crawford et al., 2018*; *Cvetković et al., 2020*).

Intra-individual variability in biological maturity is particularly relevant when analyzing changes in body composition (*e.g.*, body fat), blood pressure (BP), and cardiorespiratory fitness (CRF). This is supported by previous research showing strong associations between maturity status and fat mass accumulation (*Wang, 2002*), discrepancies between chronological and biological age in aerobic and strength capacity (*Ortega et al., 2008*), and differences in metabolic responses to exercise (*Stephens, Cole & Mahon, 2006*).

Failing to account for biological maturation may lead to an under- or overestimation of an intervention's true effectiveness, particularly in group-based analyses. For instance, early-maturing adolescents may demonstrate greater absolute gains in strength or CRF due to their advanced physical development, independent of the intervention itself. Conversely, late-maturing individuals may appear less responsive, not due to lack of training efficacy, but because of their maturational stage (*Wang, 2002*; *Stephens, Cole & Mahon, 2006*; *Ortega et al., 2008*).

In view of the non-invasive methods of biological assessment, it is quite easy to introduce this factor into the analyses. One such indicator is the predicted maturity offset (MO), defined as the time before or after peak height velocity (PHV). Calculated as chronological age (CA) at the predicted MO, it provides an estimate of age at peak height velocity (APHV) (*Mirwald et al., 2002*). Although predicted offset and APHV are increasingly used in studies of sports performance and physical activity among youth (*Malina, 2014*), they have at times been overlooked because chronological age has traditionally served as a simpler, more direct method of classifying adolescents. However, by capturing variations in biological maturation that CA alone cannot reveal, MO offers crucial insight into the timing and intensity of training adaptations across different stages of adolescent development.

Despite recommendations emphasizing the importance of individual differences in biological maturation when designing youth conditioning and training programs (*Granacher et al., 2016*; *Lloyd et al., 2014*; *Meylan et al., 2010*), many existing adolescent exercise studies focus predominantly on chronological age (CA). This leaves a gap in understanding how maturity offset (MO)—the estimated time before or after peak height velocity—might shape health-related fitness responses to intensive training. High-intensity functional training (HIFT) holds promise for improving outcomes such as body fat (BF), blood pressure (BP), and cardiorespiratory fitness (CRF), yet evidence on the modifying role of biological maturation remains limited (*Domaradzki, Koźlenia & Popowczak, 2022*; *Alvarez et al., 2017*). Therefore, the present study aimed to address this gap by (1)

investigating the effects of CA and MO on HIFT-induced changes in BF, BP, and CRF, and (2) assessing the extent to which biological maturation contributes to these outcomes beyond chronological age alone. We hypothesize that HIFT will lead to significantly greater improvements in BF, BP, and CRF among adolescents with higher biological maturity, whereas CA will be of secondary importance in explaining variance in these health-related measures.

## MATERIALS AND METHODS

### Sample size calculation

Data from 116 individuals were analyzed. After collecting these data, we performed a *post-hoc* power calculation using G* Power software to confirm the adequacy of our sample size for a general linear multidimensional analysis of variance (MANOVA) involving eight groups, two repeated measures, and their interaction term. Based on an effect size (ES) of 0.36, a significance level ($\alpha$) of 0.05, and an estimated power of 0.80, our analysis indicated that the sample size was sufficient. Because we were constrained by school conditions (*i.e.*, the necessity to randomly select entire classes without the ability to manipulate individual participants), a *post-hoc* power analysis was deemed the most appropriate approach.

### Participants

Participants' calendar age was between 15 and 18 years. They lived in the same large city of approximately 669,500 residents. Group randomization, without replacement, was carried out using the tool available at https://www.randomizer.org/. To minimize disruption to the educational process, we selected entire classes, not individual students. Each class was coded by its level (*e.g.*, 1A, 2B), and two classes from each of the four levels (1–4) were randomly chosen. These eight classes were then split, with one class from each level being randomly assigned to either the experimental (EG) or control (CG) groups. Inclusion criteria included no organized physical activity or recreational activities in the previous 6 months, no medical contraindications for physical activity, and consistent participation in PE lessons for 6 months. From those meeting these criteria, 30 individuals per age group were randomly selected and then divided into EG and CG with 15 participants each. After the project began, data from four participants (EG $n = 1$, CG $n = 3$) were excluded due to incomplete testing. No participants withdrew for any reason. All participants and their legal guardians read and signed an informed consent form approved by the Ethics Committee of the Wroclaw University of Health and Sport Sciences (No. 33/2018).

### Intervention

The experimental group performed high-intensity functional training (HIFT) during physical education (PE) lessons twice a week for 8 weeks. The HIFT circuit included 20-s intervals of squats, ab crunches, push-ups, lunges, and burpees, performed AMRAP (as many repetitions as possible), with no rest between exercises and a fixed 60-s rest after each circuit. The number of rounds increased progressively over time—starting with two rounds in the first 2 weeks and reaching five rounds in the final 2 weeks—resulting in an

**Table 1 Volume progression pattern.**

| Exercises | Week | Circles [n] | Exercise time [s] | Workout duration [s] | Rest between circles [s] |
|---|---|---|---|---|---|
| 1. Squats | 1 | 2 | 20 | 260 | 60 |
| 2. Sit-ups | 2 | 2 | | 260 | |
| 3. Push-ups | 3 | 3 | | 420 | |
| 4. Lunges | 4 | 3 | | 420 | |
| 5. Burpees | 5 | 4 | | 580 | |
| | 6 | 4 | | 580 | |
| | 7 | 5 | | 740 | |
| | 8 | 5 | | 740 | |

increase in total training time from 6 to 14 min (Table 1) (*Koźlenia et al., 2024*). This progression led to an increase in both total work and total recovery time across the intervention. Accordingly, the work-to-rest ratio decreased from 3.3:1 in weeks 1–2 to 2:1 in weeks 7–8. While the training volume increased, the inclusion of regular and sufficient rest intervals was necessary to maintain the intensity of effort and ensure proper technique and quality of movement during each AMRAP effort. These features—functional multi-joint movements, minimal intra-set rest, AMRAP structure, progressive overload, and high perceived effort—are consistent with the defining principles of HIFT (*Feito et al., 2018a*). Immediately after each HIFT workout, the rating of perceived exertion was assessed using a 0–10 RPE scale, with a target of 7–8, indicating high exercise intensity (*Duncombe et al., 2024*).

## Measurements

### Procedure

Testing sessions were conducted on two occasions: once before the 8-week intervention (baseline) and once immediately afterward (post-intervention). Each session took place in a sports hall between 8:00 a.m. and 1:00 p.m., under identical conditions for all groups. Upon arrival, participants sat quietly for 10 min before having their blood pressure measured. Next, they underwent anthropometric and body composition assessments (barefoot) while wearing T-shirts, shorts, and sports shoes. Finally, the multistage fitness test was administered. All procedures followed H-RF guidelines, with prior confirmation of their scientific justification and reliability for young participants (*Ortega et al., 2008*).

### Anthropometric measurements

An anthropometer SECA (GPM Anthropological Instruments, DKSH Ltd., Zürich, Switzerland) was used to record body height twice with 0.1 cm accuracy, following the International Society for the Advancement of Kinanthropometry protocol (*Marfell-Jones et al., 2006*). Trained staff conducted measurements to ISAK manual standards.

### Body weight and body fat

A BC analyzer (InBody230) assessed body weight and body fat using bioelectric impedance (InBody Co. Ltd., Cerritos, CA, USA). This tool is highly reliable, with intraclass

correlation coefficients for BF% (≥0.98) (*McLester et al., 2020*). Participants were instructed to refrain from engaging in physical activity and consuming food or beverages for at least 3 h prior to the measurements and to empty their bladder right before the assessment (*McLester et al., 2020*). Body mass index (BMI) was calculated using body height and weight with the formula:

Body Mass Index (BMI) $[\mathrm{kg/m^2}]$ = body mass [kg]/body height$[\mathrm{m^2}]$

### Blood pressure

All measurements were carried out with an Omron BP710 Automatic Blood Pressure Monitor. Participants were sitting quietly for 10 min before the measures. The readings were taken three times in 10-min intervals. The results introduced in the analysis were the means of the three measurements for each parameter: systolic blood pressure (SBP) and diastolic blood pressure (DBP).

### Cardiorespiratory fitness—multistage fitness test

CRF was assessed using the Multistage Fitness Test, otherwise known as the 20 m shuttle run test. This continuous sub-maximal test is useful as a field test (such as used in PE lessons) for measuring aerobic capacity. Originally developed in 1982 by *Leger & Lambert (1982)*, later modified for children and adolescents by *Leger et al. (1988)*. The test-retest reliability coefficient is 0.89, and the test result correlates with $VO_2$max (r = 0.71) (*Leger et al., 1988*). The test involves running a 20-m shuttle and reaching the opposite end of the distance before the next audible signal. The time between the two audible signals decreases each minute, requiring the participant to increase their running speed. Running velocity started at 8.5 km/h and increased successively by 0.5 km/h every minute. The results were the total distance in meters calculated from the number of stages achieved by each participant. The total number of shuttles completed by the participant was multiplied by 20 (20 = 1 × 20 m shuttle run from start-point A to cone B). For example, an individual who performed 30 shuttles received a number of 600 m (30 × 20).

### Biological maturation—maturity offset

Biological maturation was evaluated by estimating the number of years from the age at peak height velocity (APHV), referred to as "maturity offset" (MO). This estimation used a sex-specific prediction equation derived from basic anthropometric measurements (*Moore et al., 2015*). The method we utilized previously (*Domaradzki, Koźlenia & Popowczak, 2022*):

boys: maturity offset = −7.999994 + (0.0036124 × (age × body height))

Based on MO, all participants from the EG and separately from the CG were divided into four groups defined by quartiles of MO span. Hence, we distinguished four groups of matures in the EG and the same in the CG: LM—late matures (EG: MO < 2.23 years after APHV, CG: MO < 2.09 years after APHV), MLM—middle late matures (EG: MO < 3.15 years after APHV, CG: MO < 3.05 years after APHV), MEM—middle early matures (EG:

**Table 2 Descriptive statistics of the baseline demographic and anthropometrical measurements in experimental (EG) and control (CG) groups across chronological age groups.**

| Variable | EG | | | | CG | | | |
|---|---|---|---|---|---|---|---|---|
| | 15 | 16 | 17 | 18 | 15 | 16 | 17 | 18 |
| Age [y] | 15.3 ± 0.3 (15.2–15.5) | 16.9 ± 0.1 (16.9–17) | 17.5 ± 0.2 (17.3–17.6) | 18.3 ± 0.2 (18.1–18.4) | 15.4 ± 0.2 (15.3–15.5) | 16.8 ± 0.4 (16.6–17) | 17.4 ± 0.3 (17.3–17.6) | 18.3 ± 0.3 (18.1–18.5) |
| Maturity offset [y] | 1.6 ± 0.4 (1.4–1.8) | 3.0 ± 0.5 (2.7–3.2) | 3.3 ± 0.4 (3.1–3.5) | 3.8 ± 0.4 (3.6–4.1) | 1.6 ± 0.4 (1.4–1.8) | 2.8 ± 0.4 (2.5–3.0) | 3.3 ± 0.2 (3.2–3.5) | 3.9 ± 0.3 (3.7–4.1) |
| Body height [cm] | 173.4 ± 6.8 (169.6–177.1) | 179.6 ± 7.8 (175.2–183.9) | 179.6 ± 5.2 (176.7–182.5) | 179.2 ± 6.2 (175.8–182.7) | 172.5 ± 6.2 (168.9–176.1) | 178.1 ± 4.8 (175.4–180.8) | 180.4 ± 3.0 (178.8–182.1) | 180.5 ± 5 (177.3–183.6) |
| Body weight [kg] | 73.5 ± 13.5 (66–80.9) | 72.2 ± 9.9 (66.8–77.7) | 70.8 ± 6.6 (67.1–74.4) | 73.3 ± 12.5 (66.4–80.1) | 65.7 ± 9 (60.5–70.9) | 74.3 ± 10.4 (68.6–80) | 68.3 ± 5.6 (65.2–71.4) | 65.9 ± 6.9 (61.5–70.3) |
| Body fat [kg] | 15.8 ± 9.0 (10.8–20.8) | 13.2 ± 6.8 (9.4–16.9) | 11 ± 4.8 (8.4–13.7) | 12.1 ± 8.8 (7.2–16.9) | 11.8 ± 6.3 (8.2–15.5) | 14.0 ± 8.4 (9.3–18.6) | 9.9 ± 3.0 (8.2–11.6) | 7.1 ± 3.1 (5.2–9) |
| BMI [kg/m$^2$] | 24.3 ± 3.5 (22.4–26.3) | 22.4 ± 2.7 (20.9–23.9) | 21.9 ± 1.8 (21.0–22.9) | 22.8 ± 3.7 (20.7–24.8) | 22.1 ± 3.0 (20.4–23.8) | 23.5 ± 3.5 (21.5–25.4) | 21 ± 1.6 (20.1–21.8) | 20.2 ± 1.7 (19.1–21.3) |

MO < 3.59 years after APHV, CG: MO < 3.52 years after APHV), and EM—early matures (EG: MO > 3.59 years after APHV, CG: MO > 3.52 years after APHV).

### *Statistics*

Continuous variables were presented as means with standard deviations and 95% confidence intervals. Assumptions for the main analyses were examined before running statistical tests: the Shapiro-Wilk test for normality of the distribution and Levene's test for equality of variances between groups. One-way MANOVA was used to compare all groups defined by CA (EG × CG × 15-, 16-, 17-, and 18-year-olds), while MANCOVA with CA as a confounding variable was used to compare MO classes. Partial eta squared ($\eta_p^2$) was used as a measure of explained variance. Statistically significant differences were examined with ANOVA and ANCOVA (respectively), and the detailed comparisons were conducted using Bonferroni (correction) *post hoc* tests.

Multiple linear regressions were employed to estimate the predictive power of maturity status and maturity-by-years on each outcome: ΔBF, ΔSBP, ΔDBP, and ΔCRF. Semi-partial squared correlations ($r^2_{sp}$) were used as measures of effect size (*Cohen et al., 2003*).

The significance level used in all statistical tests was an α-value = 0.05, which was highlighted in bold in the accompanying tables. Statistica v. 13.0 (Statsoft Polska, Cracow, Poland) was used for all analyses.

## RESULTS

The baseline anthropometrical characteristics of the studied groups are presented in Table 2.

MANOVA showed a significant multidimensional effect of age (Wilk's Λ = 0.04, $\eta_p^2$ = 0.70, $p$ < 0.001), but no effect of the EG *vs.* CG group (Wilk's Λ = 0.94, $\eta_p^2$ = 0.06,

**Table 3 Descriptive statistics of the changes (Δ) induced by HIFT (EG) and spontaneous (CG) in body fat (BF), systolic blood pressure (SBP), diastolic blood pressure (DBP) and cardiorespiratory fitness (CRF).**

| EG-Chronological age | 15y | 16y | 17y | 18y |
|---|---|---|---|---|
| ΔBF [kg] | −0.2 ± 2.1 (−1.4 to 0.9) | −1.4 ± 1.6 (−2.3 to −0.5) | −1.1 ± 1.9 (−2.1 to −0.1) | −0.3 ± 1.1 (−0.9 to 0.3) |
| ΔSBP [mmHg] | −7.3 ± 9.1 (−12.4 to −2.3) | −3 ± 9 (−8 to 2) | −1.7 ± 8.4 (−6.3 to 3) | −10.4 ± 11.3 (−16.6 to −4.2) |
| ΔDBP [mmHg] | −4.3 ± 4.8 (−7 to −1.7) | −3.2 ± 8.6 (−8 to 1.6) | −4.1 ± 5.7 (−7.3 to −1) | −4.5 ± 5.7 (−7.7 to −1.4) |
| ΔCRF [m] | 81.3 ± 149.6 (−1.5 to 164.2) | 106.7 ± 204.5 (−6.6 to 219.9) | 157.3 ± 161 (68.2 to 246.5) | 194.7 ± 151.5 (110.8 to 278.5) |
| **CG-Chronological age** | **15y** | **16y** | **17y** | **18y** |
| ΔBF [kg] | 0.4 ± 1 (−0.2 to 1) | −0.4 ± 0.8 (−0.8 to 0.1) | 0.2 ± 1.3 (−0.5 to 0.9) | 1.2 ± 2 (−0.1 to 2.5) |
| ΔSBP [mmHg] | −0.4 ± 4.4 (−2.9 to 2.1) | −2.1 ± 3.2 (−3.8 to −0.3) | −0.8 ± 3.4 (−2.7 to 1.1) | 0 ± 4.6 (−2.9 to 2.9) |
| ΔDBP [mmHg] | −0.4 ± 3.8 (−2.5 to 1.8) | 1.3 ± 2.9 (−0.3 to 3) | 0.9 ± 2.1 (−0.3 to 2) | 0.9 ± 2.9 (−0.9 to 2.8) |
| ΔCRF [m] | 28.6 ± 146.1 (−55.8 to 112.9) | −30.7 ± 67.6 (−68.1 to 6.7) | −2.7 ± 87.8 (−51.3 to 45.9) | −48.3 ± 70.6 (−93.2 to −3.5) |
| **EG-Maturity offset class** | **LM** | **MLM** | **MEM** | **EM** |
| ΔBF [kg] | −0.4 ± 2.1 (−1.5 to 0.8) | −0.8 ± 1.7 (−1.7 to 0.1) | −1.9 ± 1.5 (−2.7 to −1) | 0 ± 1.1 (−0.7 to 0.6) |
| ΔSBP [mmHg] | −6.2 ± 10.1 (−11.8 to −0.6) | −3.1 ± 7.9 (−7.5 to 1.3) | −3.7 ± 7.6 (−7.9 to 0.5) | −9.4 ± 12.7 (−16.5 to −2.3) |
| ΔDBP [mmHg] | −3.9 ± 6 (−7.2 to −0.6) | −2.9 ± 7.1 (−6.9 to 1) | −6.1 ± 6.4 (−9.6 to −2.5) | −3.3 ± 5.4 (−6.3 to −0.4) |
| ΔCRF [m] | 53.3 ± 140.7 (−24.6 to 131.3) | 70.7 ± 148.7 (−11.7 to 153) | 122.7 ± 101.7 (66.4 to 179) | 293.3 ± 174.6 (196.6 to 390) |
| **CG-Maturity offset class** | **LM** | **MLM** | **MEM** | **EM** |
| ΔBF [kg] | 0.2 ± 0.8 (−0.2 to 0.7) | 0 ± 1.1 (−0.7 to 0.6) | 0.1 ± 1.3 (−0.7 to 0.9) | 0.9 ± 2 (−0.2 to 2.1) |
| ΔSBP [mmHg] | −0.4 ± 4.4 (−3 to 2.1) | −1.3 ± 2.5 (−2.7 to 0.2) | −0.4 ± 3 (−2.1 to 1.4) | −1.4 ± 5.3 (−4.5 to 1.6) |
| ΔDBP [mmHg] | −0.8 ± 3.7 (−2.9 to 1.4) | 1.7 ± 2.9 (0 to 3.4) | 0.8 ± 2.4 (−0.6 to 2.2) | 1.1 ± 2.4 (−0.3 to 2.5) |
| ΔCRF [m] | 38.6 ± 136.7 (−40.3 to 117.5) | −42.9 ± 73.9 (−85.6 to −0.2) | −11.4 ± 75.9 (−55.3 to 32.4) | −32.9 ± 90 (−84.8 to 19.1) |

**Note:**
LM, late matured; MLM, middle late matured; MEM, middle early matured; and EM, early matured.

$p < 0.283$). Detailed comparisons showed significant differences only in body height between the youngest groups (EG and CG 15-year-olds) in comparison to the older groups ($p < 0.05$).

Table 3 presents the characteristics of the main outcomes: ΔBF, ΔSBP, ΔDBP, and ΔCRF in the CA groups and MO quartiles. Comprehensive analysis with MANOVA showed multidimensional significant differentiation in the set of outcomes between the intervention and control groups (EGs *vs.* CGs), but not in age groups (Wilk's Λ = 0.52, $\eta_p^2 = 0.48$, $p < 0.001$; Wilk's Λ = 0.83, $\eta_p^2 = 0.07$, $p < 0.063$, respectively), except for changes in BF (F = 4.34, $p = 0.006$). ANOVA testing conducted on the intervention groups revealed that changes (Δ) induced by HIFT in EG were significantly higher than changes in the CG in all outcomes: ΔBF – F = 15.22, $p < 0.001$, ΔSBP – F = 12.19, $p < 0.001$, ΔDBP – F = 25.76, $p < 0.001$, and ΔCRF – F = 32.81, $p < 0.001$.

The next step was to assess the impact of the maturity status on changes induced by HIFT in the EG compared to the CG. MANCOVA conducted on the CGs showed no impact of the maturity status on any of the outcomes (Wilk's Λ = 0.93, $\eta_p^2 = 0.07$,

$p < 0.451$). The series of ANCOVA testing confirmed a lack of effects in each comparison ($p > 0.05$ for all outcomes).

In contrast, MANCOVA (controlling for CA) conducted on the EG showed significant differences in four outcome variable sets ($\Delta$BF, $\Delta$SBP, $\Delta$DBP, $\Delta$CRF) between all EG maturity status classes (Wilk's $\Lambda = 0.49$, $\eta_p^2 = 0.24$, $p < 0.001$). The series of individual ANCOVA tests showed significant differences between maturity classes in BF changes (F = 3.48, $p = 0.022$) and CRF (F = 8.76, $p < 0.001$), but not systolic and diastolic blood pressures (F = 1.20, $p = 0.318$; F = 0.86, $p = 0.468$, respectively). Bonferroni tests revealed a difference in BF gain only between the MEM and EM classes ($p = 0.022$; ES = 1.44). Concerning $\Delta$CRF, the trend of increasingly better values was visible. Pairwise comparisons favored EM and MEM in comparison to LM and MLM. However, statistically significant differences were seen between early and the rest of the quartiles: EM $vs.$ LM: $p < 0.001$; ES = 1.51, EM $vs.$ MLM: $p < 0.001$; ES = 1.37, and EM $vs.$ MEM: $p = 0.008$; ES = 1.19.

To more directly assess the influence of maturity status on the outcomes, we used a bubble plot (Fig. 1) to illustrate the distribution of maturity offset (MO) classes across chronological age (CA) groups. Notably, the data showed that biological maturity did not always correspond to chronological age. For instance, among 15-year-olds, 14 participants were classified in the lowest maturity category (LM), and only 1 (6.7%) fell into the intermediate-late (MLM) class. In contrast, among 16-year-olds, 8 (53.3%) were categorized as MLM, whereas 1 (6.7%) was classified as LM. The 17-year-old group displayed greater variability, with 5 (33.3%) in LM, 6 (40%) in MEM, 6 (40%) in MLM, and 1 (6.7%) in EM. For 18-year-olds, 11 (73.3%) were assigned to EM, and 4 (26.7%) to MEM. This variation indicates that while 16-year-olds spanned all four MO classes, 15-year-olds were comparatively homogeneous, and about 27% of 18-year-olds were classified as less mature. These findings informed subsequent analyses examining how maturity status impacted the study outcomes.

The results of the multiple linear regression models are presented in Table 4. Within the four outcomes, 1% to 15% of the total variance was accounted for by CA and MO. MO was the primary contributor to the variance in $\Delta$CRF ($\beta = 0.71$, $r^2_{sp} = 9.3\%$), and its load was statistically significant ($p = 0.014$). Together with the CA, both independent variables explained 15% of the variance changes in CRF induced by HIFT, and the model was significant ($p = 0.004$). The influence of CA and MO was very similar and statistically insignificant in the rest of the outcomes.

## DISCUSSION

This study investigated the effects of CA and MO on the magnitude of change in key H-RF outcomes ($\Delta$BF, $\Delta$SBP, $\Delta$DBP, and $\Delta$CRF) after 8 weeks of a HIFT intervention during PE classes in male adolescents. In addition, the study also assessed the role of maturity status alongside CA in individuals' improvements in these outcomes after HIFT. In general, we found that MO was the primary determinant of improvement in CRF and had a smaller

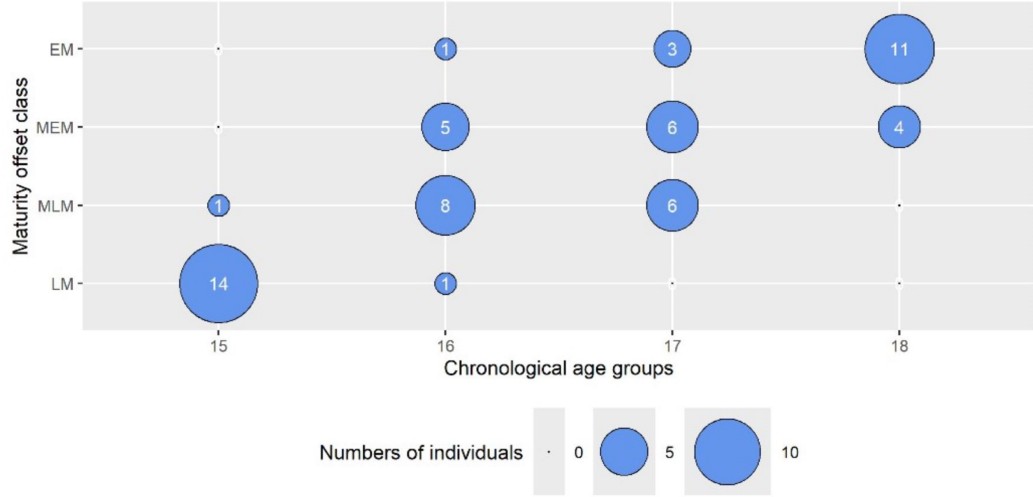

**Figure 1 The bubble plot presents the numbers of individuals from the chronological age groups in maturity offset class categories.** The wheel size represents the number of individuals. LM, late matured (biologically youngest, maturity offset <2.23 years after APHV); MLM, middle late matured (maturity offset <3.15 years after APHV); MEM, middle early matured (maturity offset <3.59 years after APHV); and EM, early matured (biologically oldest, maturity offset >3.59 year after APHV).

**Table 4 Multiple linear regressions with chronological age (Age) and maturity status (Maturity offset) as outcomes predictors.**

| Variable | Predictor | β | $p(\beta)$ | $r^2_{sp}$ | $R^2$ | $p$ |
|---|---|---|---|---|---|---|
| ΔBF | Age (y) | −0.22 | 0.478 | 0.009 | 0.02 | 0.710 |
| | Maturity offset (y) | 0.14 | 0.650 | 0.004 | | |
| ΔSBP | Age (y) | −0.02 | 0.952 | 0.001 | 0.03 | 0.917 |
| | Maturity offset (y) | −0.04 | 0.902 | 0.001 | | |
| ΔDBP | Age (y) | 0.40 | 0.189 | 0.030 | 0.01 | 0.368 |
| | Maturity offset (y) | −0.43 | 0.160 | 0.034 | | |
| ΔCRF | Age (y) | −0.34 | 0.222 | 0.022 | 0.15 | 0.004 |
| | Maturity offset (y) | 0.71 | 0.014 | 0.093 | | |

Note:
ΔBF, change in body fat; ΔSBP, change in systolic blood pressure; ΔDBP, change in diastolic blood pressure; ΔCRF, change in cardiorespiratory fitness; β, standardized beta coefficient; $p(\beta)$, $p$-value for β; $R^2$, full model squared correlation; $r^2_{sp}$, semi partial squared correlation.

effect on ΔBF. In contrast, CA had an insignificant effect on ΔBF. Neither maturity status nor CA influenced changes in SBP or DBP.

Calendar age and biological maturation are critical factors influencing a youth's morphological development and functional capabilities, as they drive the growth processes that underpin differences in body structure and function (*Ortega et al., 2008*). Despite all age categories responding positively to HIFT, older boys showed a significantly greater increase in CRF—an average improvement of 194.7 ± 151.5 m. This enhanced response

may be attributed to advanced neuromuscular development, greater training tolerance, and higher baseline muscle mass typically observed in older adolescents. However, it is important to acknowledge that measurements of CRF, particularly in field-based assessments such as the 20-m shuttle run, may be influenced by individual pacing strategies, levels of motivation, and familiarity with the test protocol, which could partially contribute to the observed variability in performance outcomes. A study by *Gillen et al. (2019)* shows that neuromuscular adaptations appear to play a significant role in strength development from pre-adolescence through adolescence. For instance, in our sample, older participants demonstrated higher baseline lean body mass and stronger handgrip measures (both $p < 0.05$) than their younger counterparts. For instance, in our sample, older participants demonstrated higher baseline lean body mass and stronger handgrip measures (both $p < 0.05$) than their younger counterparts, suggesting a more advanced neuromuscular profile. This maturational advantage may partly explain their greater responsiveness to the HIFT intervention and highlights the importance of considering biological development when designing age-appropriate training programs in school settings. Coupled with age-related improvements in motor unit recruitment and coordination, older adolescents may therefore demonstrate greater adaptability to high-intensity functional training, resulting in more pronounced physiological benefits. This suggests that biological maturation enhances not only exercise tolerance but also the effectiveness of training interventions in this age group. In contrast, younger boys still displayed meaningful improvements, but their smaller gains underline the role of maturity-related factors in driving differential training adaptations across age groups. Age-related differences in CRF improvements following HIFT have been reported, yet some research suggests these adaptations may not rely specifically only on biological maturation. For instance, *Alvarez et al. (2017)* showed that exercise outcomes can occur regardless of a youth's developmental stage, while *Brisebois, Rigby & Nichols (2018)* found that maturation alone does not decisively govern CRF gains. In practice, older adolescents often benefit from factors like advanced neuromuscular development and higher baseline fitness, which can amplify training responses. However, individual variability in growth trajectories means that relying solely on chronological age or a single maturity assessment may overlook subtle but important differences in how adolescents adapt to training. Consequently, a more tailored HIFT approach—one that accounts for both age-related and individual maturation factors—could better optimize CRF improvements for each participant. Although prior research has recognized the morphological and neurological advantages held by more mature adolescents, the precise impact of biological age on short, intensive intervention programs remains unclear. Indeed, these individuals often display superior physical performance across multiple fitness domains. For instance, in a study of 904 female basketball players (U13–15), those at a more advanced maturity stage (PHV3) outperformed less mature peers in 10 and 20 m sprints, jumping, agility, and endurance tests ($p < 0.05$), even when controlling for age, height, and body mass (*Gryko et al., 2022*). These differences suggest that advanced neuromuscular development—reflected by better sprint times and jumping abilities—significantly contributes to higher overall fitness in more biologically mature adolescents. However, additional data are needed to fully
examine how maturation status influences outcomes from brief, high-intensity interventions in school-age populations (*Domaradzki, Koźlenia & Popowczak, 2022*). Further addressing these gaps could help determine whether age-related biological factors produce meaningful differences in intervention effectiveness, ultimately guiding the design of age-appropriate training protocols.

Previous research (*Domaradzki, Koźlenia & Popowczak, 2023*) showed varied responses to short intensive intervention programs, suggesting that individual characteristics may influence outcomes. Further studies are required. Our current findings extend these observations by demonstrating that biological maturation significantly moderates the improvements in cardiorespiratory fitness (CRF) following HIIT. Specifically, individuals at more advanced stages of maturation exhibited greater CRF enhancements—likely driven by underlying hormonal changes (*e.g.*, increased testosterone, growth hormone, and IGF-1) that stimulate lean tissue accretion and neuromuscular development, which in turn improve oxygen transport and muscular endurance. These maturational adaptations may foster a higher training tolerance and faster recovery, thereby enabling more pronounced gains from high-intensity training compared to less mature peers. However, the relationship between advanced maturity and body composition is multifaceted: a recent study of Azorean youth reported that youth with advanced maturation status were also at higher risk of overweight/obesity and low CRF (*Coelho et al., 2013*). This highlights that, although maturational progression can offer physiological advantages that enhance CRF under sufficient physical activity, it may also predispose adolescents to excess weight gain if energy intake exceeds expenditure or physical activity remains inadequate. Therefore, while biologically mature adolescents often see amplified benefits from high-intensity interventions, it is vital to incorporate systematic monitoring of maturation status into physical activity programs to balance the potential for greater fitness gains with the heightened susceptibility to weight-related issues. Considering the observed variations in HIIT's impact on CRF due to biological maturation, it is imperative to incorporate maturation status into the design of physical activity programs. Studies indicate that rapid maturation is linked with enhanced physical performance (*Scalzo et al., 2014*), supporting a mechanistic explanation where accelerated hormonal changes facilitate neuromuscular adaptations that enhance aerobic capacity. It is also noted that individuals respond differently to physical training loads (*Kibler et al., 1989*), and baseline fitness levels further modulate these responses (*Bogataj et al., 2021*).

Short, intensive efforts can significantly enhance CRF among adolescents. For instance, our own study (*Domaradzki et al., 2020*) showed a significant improvement in Physical Efficiency Index (PEI), with underweight male participants increasing from 41.63 ± 4.54 to 45.72 ± 2.93 ($p < 0.05$). Similarly, an overweight group improved from 41.45 ± 4.16 to 46.70 ± 3.93 ($p < 0.05$). In another study, *Bogataj et al. (2021)* observed an ~8.8% increase in Yo-Yo Intermittent Recovery Test Level 1 (YYIRT1) performance (1,202.73 ± 143.9 m to 1,308.18 ± 79.68 m), though the group effect did not reach statistical significance ($p = 0.438$, $\eta^2 = 0.013$).

Collectively, these findings underscore the broad applicability of brief, high-intensity protocols—often classified as HIFT or HIIT—for improving aerobic capacity and body composition in youth populations, suggesting that even time-efficient training formats can confer meaningful health and performance benefits (*Arrieta-Leandro, Hernández-Elizondo & Jiménez-Díaz, 2023*). The integration of HIFT into physical education lessons is particularly promising in counteracting global trends of decreased physical activity (*Tjønna et al., 2009*). This approach not only improves CRF but also contributes to metabolic health and overall physical fitness in adolescents.

The efficacy of high-intensity intermittent efforts has been corroborated by *Batacan et al. (2017)* and a meta-analysis by *Solera-Martinez et al. (2021)*, while studies among young athletes and school-aged children (*Engel et al., 2018*; *Cvetković et al., 2018*) further confirm HIIT's broad benefits. *Hsieh et al. (2014)* and *Bonney, Ferguson & Smits-Engelsman (2018)* have linked increased body mass with reduced CRF, yet our analysis did not reveal a significant impact of adiposity on CRF outcomes. This discrepancy suggests the existence of compensatory physiological mechanisms—potentially involving metabolic flexibility—that mitigate the negative effects of higher body mass. Additionally, research by *Ouerghi et al. (2017)* and *Lambrick et al. (2016)* supports the substantial benefits of HIIT for individuals across different weight categories. Collectively, these data advocate for a personalized approach to training program design that accounts for individual baseline characteristics, such as maturation status, to optimize both CRF and metabolic health outcomes.

The present study is not without limitations. First, narrowing age groups to secondary school limited a greater span of MO and made a bias toward matured individuals. We collected individuals after APHV. The inclusion of primary school students would provide a greater range of MO, including participants before the APHV. The second limitation is linked to sample specificity, *i.e.*, only male participants. Thus, we did not observe trends in females. The third limitation refers to the method used to assess biological maturation and the technique used to classify adolescents into different maturity groups. The MO was calculated using the predicting equation published by *Moore et al. (2015)*. There is no sitting height in this formula. Despite very high reliability, other formulas are more precise. Moreover, most authors use formulas with sitting height included, which makes easier comparisons in the discussion. Due to the organization of the school process, we were not able to randomize students individually, so we had to randomize entire classes. We did not control dietary intake. Although validated equipment and standardized procedures were used for blood pressure measurement, potential variability due to individual physiological fluctuations, time of day, or emotional state cannot be fully excluded. Future studies may benefit from conducting repeated assessments across multiple days to enhance reliability.

The present study also has a number of strengths. Firstly, the set of variables used in our study allows for exceptional multivariate analyses. Moreover, this set consists of H-RF outcomes that are very important for an adolescent's health assessment. Secondly, combining two factors in the analyses (CA *vs*. MO) allowed us to evaluate a concurrent

approach to the classification of respondents. Thirdly, the maturity status was revealed as a strong determinant in the benefits of HIFT intervention on CRF. So, it should be in the sights of teachers who decide to introduce interventions with short efforts into PE lessons. Fourthly, our findings are expected to have implications for authorities and decision-makers who are responsible for school programs. They should consider making it mandatory for teachers to include short, intensive intervals systematically and frequently in every PE lesson.

## CONCLUSIONS

Maturity status was linked to the amount of post-intervention benefits in CRF in male adolescents in contrast to the rest of the outcomes. Therefore, our hypothesis was partially fulfilled due to other factors that were not affected significantly. When planning to include HIFT interventions in PE lessons, teachers should be aware that maturity status is acting as a determinant in the CRF gain, which could explain results differentiation between individuals. Our results offer valuable insights for practitioners by highlighting the effects of a bodyweight intervention, based on the HIFT concept, performed during physical education classes on CRF. This study introduces a novel approach for educators, enabling them to adopt a more tailored strategy to help adolescents enhance CRF. Future studies should also aim to examine the influence of maturity status in the long term, *e.g.*, along a semester, as well as to examine the follow-up effects. Moreover, this type of research should also focus on female adolescents, as well as younger students in the pre-pubertal period.

## ACKNOWLEDGEMENTS

We would like to thank the employees of the Agnieszka Osiecka Secondary School No. XVII in Wrocław, Poland, for their participation in this research.

### Funding
The authors received no funding for this work

### Competing Interests
The authors declare that they have no competing interests.

### Author Contributions
- Jarosław Domaradzki conceived and designed the experiments, analyzed the data, prepared figures and/or tables, authored or reviewed drafts of the article, and approved the final draft.
- Cristian Alvarez analyzed the data, authored or reviewed drafts of the article, and approved the final draft.
- Rafał Szafraniec performed the experiments, analyzed the data, authored or reviewed drafts of the article, and approved the final draft.
- Dawid Koźlenia performed the experiments, analyzed the data, prepared figures and/or tables, authored or reviewed drafts of the article, and approved the final draft.

## Human Ethics

The following information was supplied relating to ethical approvals (*i.e.*, approving body and any reference numbers):

The Senate Ethics Committee of the Wroclaw University of Health and Sport Sciences (Ethical Application Ref: No. 33/2018).

## Data Availability

The raw measurements are available in the Supplemental File.

## Supplemental Information

Supplemental information for this article can be found online at http://dx.doi.org/10.7717/peerj.19756#supplemental-information.

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
