# Peer review of "Biological maturation determines the beneficial effects of high-intensity functional training on cardiorespiratory fitness in male adolescents"

_PeerJ, doi:10.7717/peerj.19756_

## Round 0.1 · original submission · Major Revisions

The reviewers have comprehensively evaluated your manuscript, highlighting significant improvements needed in the experimental design and reporting. We look forward to your response.

Reviewer 1 ·

Basic reporting

Thank you for inviting me to review this article. Overall, the study is well written, nevertheless, not provide original data and the design is not very comprehensive. It's a salami type study.

Introduction
L46: In the first paragraph, you mix the sedentary behavior of the adolescents, the effects of exercise and aspects of interval training. Please present in different paragraphs. Also, the use of the term “high intensity interval training” is redundant, given that if the regime is interval training it can already be deduced that it is performed at high intensity.
L48: Include meta-analysis outcomes:
https://doi.org/10.1139/apnm-2023-0329
https://doi.org/10.1111/sms.14133
https://doi.org/10.1136/bjsports-2013-092576
https://doi.org/10.1136/bjsports-2015-095841
L58: All the training modalities are adaptable to training status.
L60: Include meta-analysis outcomes:
https://doi.org/10.3390/ijerph19159559
L64: Include this study:
https://doi.org/10.1177/00315125221083180

L71: Include current data about sedentary lifestyle in adolescents.
L101: What is the hypothesis?

Experimental design

Materials & Methods
Describe the type of design.
L115: What type of medical contraindications?
L132: Round or circuit?
L147: Validation of the instrument. Please describe more about BP measurements.
L165: You estimated the VO2max?
L199: You used Cohen d for post hoc test?

Validity of the findings

Results
Too long. Please present the results using tables and figures.

Discussion
What is the innovation? Is the hypothesis fulfilled? What are the practical applications?
L280: Search with other keywords, for example circuit training, body weight training, etc.

Reviewer 2 ·

Basic reporting

Abstract and Manuscript Structure
Comment 1 – Is this an experimental study? I suggest the authors use ‘casi-experimental’. The authors can't control all the variables and state this factor in the limitations section. The authors only claim to have randomised. However, they do not explain the methodology of the randomisation process.
Comment 2 - The abbreviations HIFT and BF must be described before they are used. This error occurs in the abstract and throughout the introduction.
Comment 3- For readers' better understanding, the authors should explain the different organisations of high-intensity interval training (HIIT) (e.g. Short, Long,....) as well as the various physiological requirements for a workout to be considered HIIT.

Experimental design

Comment 4 – How did the authors control the intensity of the training and ensure that the participants respected the physiological intervals of HIIT?
Comment 5 - Can you explain how you randomised the sample?


Comment 6 – The authors used 116 participants. They don't explain why they used 116 participants. Identify the calculations to analyse the power sample.

Comment7 – The authors used bioimpedance to calculate fat mass, but did not identify the requirements or inclusion criteria for carrying out the bioimpedance assessment. How do they ensure that the participants met the bioimpedance requirements? Can you explain the choice of obtaining fat mass values by bioimpedance instead of another method?

Comment 8– The authors performed strength exercises during their intervention. However, the main outcome of this study is cardiorespiratory fitness. According to the authors, do they consider the physical fitness test to be the most specific for assessing the functional effects of this intervention? Why?

Validity of the findings

Comment 9– The authors talk about body fat, but they don't give the baseline values in the descriptive table. Were the mean and CI values for BF the same for all groups? How can BF affect the results of the intervention?

Comment 10 – “Thirdly, the maturity status was revealed as a strong determinant in the benefits of HIFT intervention on CRF”. Body composition (fat mass) can be a confounding variable in the aerobic response. Do all the groups have the same average BF?

---

## Round 0.2 · Major Revisions

Dr. Szafraniec, the Reviewer has identified a number of issues which must be addressed in all sections of your manuscript. Please address each of the issues at your earliest convenience via a point-by-point response.

Thanks, A/Prof M Climstein

·

Basic reporting

While the overall idea is there beneath the surface, the current submission could make significant improvement in the clarity and depth of discussion.

First and foremost, clarity is needed to better bridge the gap between physical inactivity, HIFT and maturation/age and establish the rationale of the work. With a clearer rationale in place, the hypothesis provided will appear more fitting and can be carried through the entire piece.

In the discussion, these same ideas should be more clearly addressed using the findings in this paper and with critical discussion around the work of others. The discussion left me without a resolution, still thinking so what?

The methods, whilst featuring much of the correct information, potentially require rejigging and would definitely benefit from more clear, concise description. For particular value, authors should seek to outline the reliability or 'source' of various methodological components - addressing each in a systematic way. Somewhat explained by the large amount of abbreviations, it was often difficult to follow the methods point-to-point.

The result also fall victim to the same issue - the ideas are there, but are hard to digest in a flowing manner. Consistent formatting of the figures and use of descriptive data to give context to the result would be very useful.

Regarding the references, the authors have done well to include a reasonable quantity of primary research but often, have failed to expand on what the studies measured, who they measured and/or the results generated. As a result, the references barely scratch the surface, leaving most sections (particularly the introduction and discussion) wanting.

Experimental design

The experimental design appears mostly relevant to the question at hand. The question requires a more clearer, flowing outline - better moving form the general opening point re: physical inactivity to the measurement of various anthropometrics and CRF. What? Why? How?

See '1' and '4' for comments re: methods.

Validity of the findings

Novel, potentially impactful data - although the latter requires more areful consideration and discussion.

Physical inactivity yes - HIFT ok - 15-18 year olds and maturation... so what? Why here?

See '1' and '4' for pointers towards a better conclusion.

Additional comments

> Abstract <

The aim for the current study appears valid (abstract), but lacks a self-contained and concise description of the reason for the work. What information can you give to explain the changes in body fat, blood pressure and CRF expected as someone ages between 15-18 years?

In the abstract, it may also be useful to explain maturity-offset (MO) briefly, if possible. Furthermore, what about specifying the characteristics of the various groups as it relates to MO... in this way, what is BH? Body height? Please try to outline in full before abbreviating.

While I appreciate the limited word count may make it difficult, would it be possible to give some idea of what the 8 weeks of HIFT entailed? I would be interested in the intensity/duration/frequency of the sessions?

Beep test or bleep?... 'Collected using standard procedures' doesn't really mean anything. Either summarise these 'standard procedures' or don't say anything at all.

In a similar light to the above comment, 'compared using specific statistical procedures' doesn't really mean anything. At least give some kind of idea what these procedures might be... if not, remove the non-sentence. If the following sentence to this re: MANCOVA/ANCOVA is immediately relevant, you need to tie these sentences together better.

Re: the sentence first reporting the output from the Wilks lambda, replace the semi-colon before respectively with a comma.

When reporting the findings re: MO (results) as the 'main determinant' - is this relative to chronological age? If so, perhaps it could be made more clear.

Im not sure "matures" is a valid descriptor of young groups of males with different MO. Perhaps 'those who matured earliest (and then give their MO info)' would be more descriptive than "early matures".

When you say 'early matures (biologically older) benefitted more from HIFT than late matures' - be specific... how so? Descriptive/inferential statistics?'

The conclusion of the abstract appears valid but maybe lacks an answer to the question - so what? I can probably agree with what you have said in this part but so what? How can you generalise this for the next person to pick up from? Or... whats next?

> Introduction <

(L46) It would probably be useful to define physical inactivity and/or what is meant by 'is still growing' - from what to where?

(L47) I would accept the point that there is a link between insufficient physical activity (inactivity even) and adolescent obesity, increased BF and decreased CRF, but it might be useful to expand on the what and why or, at least give some data linking the two.

(L50) It might be useful to define high-intensity exercises. It would be preferable that this explained what this means physiologically, but in one way or another, be specific.

(L50) Im sure meta-analyses have shown HIIT to improve CRF in different populations but try to be specific. Measured how? Typical HIIT 'intervention'? To what extent were improvements observed?

(L52-56) Before moving on to discussion of the definition of HIIT, answer the question 'so what?'. HIIIT does all these wonderful things but, so what? Link back to your point re: physical inactivity before moving on...

(L57-59) This definition would probably be more useful much earlier on. Either way, be specific on what you mean by each of the domains or thresholds. My preference would be a physiological measure i..e., HR or relative to VO2max... The former may tie nicely into the Duncombe reference (L61-62).

(L63-75) Im not really sure what the value of this section is? It seem a little off on a tangent if compared to where we started and/or maturation. As a minimum, it would be a good idea to get to the discussion of younger individuals and CRF/PA. In this part - perhaps it would be a good idea to try and avoid talking about children or adolescents too much, given this covers those potentially as young as 10 (or less) rather than those closer to those aged 15-18 years old, as measured here.

(L78-82) So what? Why?... Why do this group not engage with PA? What is the significance to the current study?

(L87...) This is definitely relevant discussion but in this section (introduction), it feels like we have just jumped to this point without obvious, flowing progression. Think how you can better link physical inactivity/HIFT and maturation together...

It seems you have just discussed physical inactivity and the potential issues. If this is the problem, why are you not discussing methods to increase PA time instead of maturation? Be more clear on why the latter is of interest.

(L111-114) he hypotheses presented are probably quite reasonable but, I do not think your introduction has outlined the case sufficiently.

> Materials & Methods <

(L116-117) I think this sentence says a whole load of nothing. Yes it is a QE design... what is meant by the mention of the previous work?

(L122) It is good that you performed a power calculation, but it would be even better if you specified the sample size established. The way this is presented suggests that you collected data and then worked out how many people were needed.. this might still be accurate, but still needs clarification.

(L124) It may be more useful to specify the mean ± SD for age. Simultaneously, be specific on sociocultural backgrounds - what does this mean? relevance?

(L123-143) This portion appears to include the relevant information, finishing in the right place by specifying the final participant group. Could the key information be retained in this section, but instead be written in half the words?

(L148-151) Some good detail.. is this protocol based on any previous research? I am particularly curious on the attention given to ensure this was 'high-intensity' or at least the rationale.

(L153-155) How was RPE used to monitor training load? Session RPE? If so, please expand for the reader.

Furthermore, what is meant by the point re: 7-8 being considered reliable? What does one's perception of effort have to do with reliability? If there is some underlying reliability measure I am missing, provide the statistics.

(L158) Why not just specify stadiometer or the exact anthropometer tool used?

(L164-165) Please try to report the reliability statistics for this tool/method.

(L166-168) Could you possibly provide a reference for this?

(L171-178) Purely preference, but perhaps this could be said in fewer words?

(L179-190) This section is good and reasonably clear. To improve, maybe it would be good to comment on the reliability or use of this measure as an indicator of CRF? You could also ask, why not look at VO2 max or another alternative?

(L192-203) Purely my preference, but perhaps it would make sense to establish the procedure/running order first and then give details, while avoiding repetition?

(L210-216) Is this organisation of groups based on a previous definition or study that you may reference?

> Results <

Table 2... Can we maximise the font size within the current limits (consider rotating the table landscape, from top to bottom of the page)? Please also remove the vertical lines completely and minimise the use of horizontal lines.

(L239) Perhaps you could specify whether you mean age group/maturation group or other - what is 'the group'?

(L239-241) Please try to improve the clarity of this sentence... are you referring to anthropometrics and only BH was sig different? Either way, please also provide the descriptive statistics (not just the p value).

Table 3... The formatting of this table is much improved from table 2 (Minimal horizontal lines, no vertical lines). Try to ensure all all formatted consistently.

(L243-244) What is meant by the following? 'Showed multidimensional significant differentiation in the set of outcomes'... this doesn't really say anything.

(L248) What is meant by 'on higher level'... please clarify or better integrate with actual descriptive/inferential statistics.

(L251) Referring particularly to the start of this sentence , stick to scientific language and delivery. If you want to establish some separation, perhaps introduce some subheadings?

(L256) What is meant by the following? 'showed significant differences in four outcome variable sets'.

(L267-280) Im not really sure what the overall point is here. Plus, it is written in occasionally non-scientific terms.

> Discussion <

(L291-296) I would accept the first part of the discussion, IF the results were presented in a manner that was a tad easier to decipher.

(L297-304) This section doesn't say a great deal of anything. My preference would be that you use your actual results to establish improvements in some of the 'H-RF' outcomes and then critically discuss existing research, where possible.

(L302) What is the HIFT conception?

(L303-304) If HIFT is similar to HIIT, which is well reported on, why can't you discuss the results?

(L305-206) Give the specific data to reaffirm support (and give context to) this point. At some point you might also add the why.

(L306-311) I am not really sure on your point... You have stated that age effects change in CRF. Then that this could be independent of maturation... what? why? how? In a third point you have then stated something seemingly unrelated/unsupported about maturation trajectories? The final point then talks about tailored approaches again, why?

Perhaps you need to take more time to fully expand on these points before drawing them together.

(L312-313) Ok, why?

L316-317) How can you bridge the previous point to those talking about school-age populations?

(L318) No further studies are warranted?

(L318-349) I am not really sure how the discussion here is a) explaining your results and b) tidying up the overall argument of the project...

As a minimum, you need to think about the flow and relevance of each point. From there, try to avoid exclusively making broad general statements and try to dig deeper and/or provide actual data to explain your point. Some mechanistic explanation might also be good to go beyond the surface description of 'increases and decreases'.

---

## Round 0.3 · Major Revisions

Authors, you have not satisfactorily addressed the issues and concerns by the reviewer. Please refer to the Reviewer 3 comments and ensure you fully address all of the points raised.

·

Basic reporting

As evident in section (4) of my review, my main criticisms of this piece in terms of basic reporting can be broken down as follows:

1. Clarity and conciseness. Can you give all the detail in a much more digestible manner.

More importantly:

2. Focus more on the 'why'. There are many cases where you have successfully included a reference, but not provided any detail. I am sure the reference supports your point, but we need a summary of the findings. Think about discussing mechanisms too. Not just 'what' but again 'why'? Including values can tick this box in most cases.

Experimental design

Appears valid. Some minor housekeeping points (4) and a general polishing of the writing (conciseness and flow) would be sufficient here.

Validity of the findings

Appears relevant. Referring back to (1), the general ideas and information appear reasonable and valid, but could be delivered in a much more effective, detailed way. Imagine that at present you have shown me all the parts, but not what they do and how they link.

Additional comments

Please find below a range of specific comments that I believe will bring the piece closer (again) to the required standard.


(L30) Please make sure that abbreviations are fully detailed on first use. I assume ‘CA’ refers to chronological age, but you have not specified this.

(L39-41) Please amend to ‘delta’ alone as the phrase ‘delta changes’ appears redundant.

(L44-45) In the final sentence of the results here, I am fairly confident you are referring once again to maturity offset …‘together with CA’… but please be explicit to improve clarity.

(L48) Unrelated appears to be an overstatement, perhaps ‘were not significantly related’ would be more accurate?

(L49-L53) The close of the conclusion here appears a reasonable and valid point but could perhaps be more concise. With the newly gained words, you may also briefly comment on what you mean by tailoring of interventions… how could this actually be achieved?

(L57-59) A separate reference is needed to support this claim re: prevalence. Can you find a reference or multiple references that support this? Be explicit on the where also, if possible.

(L59-63) I don’t doubt that they are, but please reference some examples and be precise when detailing time. What is the time in some? Are these sessions too short or sufficient for a single bout but not weekly PA?

(L63-67) This is a fair point, but can you make it a tad more concise? For example… ≥ 85% of VO2max, >80% or age-predicted maximal heart rate and subjectively rated at least 8 on a 10-point rating or perceived exertion (REP) scale.

(L67-68) A number may be better here as increasingly could mean anything from minutes to hours a week. If this is not possible, try to describe what you mean by increasingly in some other way.

(L69-73) I would get rid of the point re: various definitions at the start (if you are not going to expand upon it) and instead, just add a note at the end that outlining that ‘definitions may vary’. Or simply use the word ‘may’. You may also amend this paragraph to better link into the mention of HIFT. For example, definitions are hard because variations exist… HIFT.

(L74-85) Here and throughout, please try to introduce data or refer to some form of values. For example, you mention ‘higher levels of enjoyment’… Measured how? How much higher? Why?

(L86) I think you need a bridging point before starting here. Why are we now talking about H-RF outcomes? What is the link between the last thing you said?

(L96-97) Would it be possible to comment on why it has been overlooked? (MO)

(L102) What is APHV? You have outlined PHV but not this. Please also clarify what you mean by ‘increasingly’… give a number or at least seek multiple references.

(L103-115) If the aim of an introduction is to justify the aim (rationale) then I think there is still work to do to more clearly tie together your discussion and then slot your project in logically. The ideas are there but need brought together better really. What? Why? How?

(L124) It may be useful to give the precise number to substantiate this.

(L139-140) This is a bit of a non-point; I would personally just remove it.

(L147-151) Given your earlier point on how you might describe HIIT (or HIFT)... it might be useful to comment on the approach to HIFT here. Did you take any measures yourself to ensure it was ‘HIFT’? Work to rest ratio may be useful.

(L166) What about the concept of BEI as a whole though? Is that reliable? Perhaps comment on this at some point, even if only as a small limitation.

(L173-179) Make company make/model? Refer back to L164 where you have given the required information. For the procedure as a whole, I would significantly condense the outline of what represents a fairly standard procedure.

(L181) Beep or bleep?

(L194) I personally think it would make sense to generally outline the procedure as a whole before giving detail on the individual components.

(L302) What is meant by ‘HIFT conception’?

(L303) Directly is not spelled correctly.

(L310-314) Give some data and/or expand on these points. For example, you have mentioned neuromuscular development, but it would be great to add more. Measured how? Differences in values? This general comment could probably be applied from top to bottom of this piece.

(L329) Avoid sweeping statements only and add the values or expand. What can you summarise from this reference (or any others) that would avoid me having to seek all the detail separately?
(
L339-342) This point could be one of the most valuable in the discussion. Please take the time to expand on the mechanisms here and throughout. There is a lot of ‘who’ and ‘what’ but not much ‘why’?

(L349-352) This details the point above and repeated throughout the whole piece. Avoid just giving references without context… Point, Evidence and Explain.

---

## Round 0.4 · Major Revisions

Authors, The reviewer has acknowledged that your manuscript is progressing in the right direction; however, has raised several key concerns that must be addressed before further consideration. Specifically, noted that the rationale remains unclear, the Introduction and Discussion sections lack coherence, and there is a need for improved clarity and specificity throughout. The reviewer has provided detailed feedback, particularly regarding the flow and justification of the Introduction and the interpretive depth of the Discussion. Please carefully address all points raised to strengthen the manuscript. Thanks, A/Prof Mike Climstein

·

Basic reporting

See additional comments and previous reviews for my general points here. Still too many points that leave the question ‘so what, why?’ unanswered.

Experimental design

Remains appropriate but could be better justified.

Validity of the findings

The work appears largely valid and I am unable to critique the method/results any further. It appears the rationale for the work and explanation of the outcomes are where this paper is most lacking.

Additional comments

Specific comments:

Introduction:

(L73 to 83) The additions here are repetitive of the opening sentence and collectively this point re: recommendations could be condensed into half as many sentences.

(L78...) The use of HI is a little repetitive, perhaps you can transition better to discussion of HIIT. In my personal opinion, it would make sense to introduce the use of HIIT after the opening discussion of WHO guidelines and then work in some form of definition for HI.

(L80-83) The point here detailing proposed improvements requires more explanation. What improvements specifically? Supporting data?

(L83-84) Capturing my second point re: L78, I would try to avoid jumping between terms. If your aim is to talk about HIFT… could you not pass over HIIT to some degree? This is a little cumbersome otherwise.

(L86...) ‘Increases motor recruitment more than x’ requires further explanation and data. Measured how? Values?

(L89-91) It makes sense that you would also provide values for this proposed enjoyment, particularly as the previous changes have been quantified briefly.

(L98-102) I am not entirely sure it is clear how we have now got to talking about maturation. Perhaps clarify why there would be any focus on maturation, if the aim is more to demonstrate adherence towards WHO recommendations. If there are improvements, I am not sure why it matters that some will vary? Be clear.

(L98-117) Repeating the above… I am not entirely convinced how we have come to focus on maturation. Be clear on how we have gotten from recommendations around activity to here. It seems to be two potentially unrelated ideas that need relating.

Discussion:

As previously mentioned, the discussion reads a little like a list of reasonable points, not a cohesive and progressive explanation of the results. When reading I am left asking for more clarity and questioning where things are going. Here are some example points within that idea:

(L315-318) What is the point you are trying to make here? IF it is to outline that it has a wide range of benefits, I think this could be made clearer. It also seems odd to make a general point around function and morphology but then name physiological systems… I would prefer a tad more specificity that you can now comment towards in this section.

(L318-319) As above, be specific and/or follow up these statements with your own data… discuss.

(L322-324) Avoid the comment re: not discussing the results. Focus on what you can explain and explain the significance of where you cannot, only when necessary.

(L328-329) It is probably worth commenting at some stage (not necessarily this section) that there is some degree of variation associated with measuring CRF this way, consider varying motivation/pacing as obvious explanations.

(L333-335) 'This is especially evident when considering major force-generating muscles and the substantial strength variations that are more closely linked to biological maturation than to chronological age’... I am not entirely sure there is a point here.

(L337-338) ‘Suggesting more mature neuromuscular status and superior force-generating capacity’… this seems repetitive of the actual result presented but how does it contribute to a bigger picture? How could we use this?

(L339-340) ‘Thus be better equipped to handle the demands of high-intensity functional exercises’… is there a difference between varying responses and ‘handling’ HIFT in general. Again, what does this point contribute to the bigger picture?

---

## Round 0.5 · accepted · Accept

Authors, thank you for addressing all of the reviewers' comments. I am pleased to recommend your amended manuscript for publication. Thank you for choosing PeerJ. We look forward to receiving future manuscripts from your research team